# IGF2BP1—An Oncofetal RNA-Binding Protein Fuels Tumor Virus Propagation

**DOI:** 10.3390/v15071431

**Published:** 2023-06-24

**Authors:** Markus Glaß, Stefan Hüttelmaier

**Affiliations:** Institute of Molecular Medicine, Martin Luther University Halle-Wittenberg, Kurt-Mothes-Str. 3a, 06120 Halle, Germany; stefan.huettelmaier@medizin.uni-halle.de

**Keywords:** IGF2BP1, virus, HBV, HCV, HPV, HIV, SARS-CoV-2

## Abstract

The oncofetal RNA-binding protein IGF2BP1 has been reported to be a driver of tumor progression in a multitude of cancer entities. Its main function is the stabilization of target transcripts by shielding these from miRNA-mediated degradation. However, there is growing evidence that several virus species recruit IGF2BP1 to promote their propagation. In particular, tumor-promoting viruses, such as hepatitis B/C and human papillomaviruses, benefit from IGF2BP1. Moreover, recent evidence suggests that non-oncogenic viruses, such as SARS-CoV-2, also take advantage of IGF2BP1. The only virus inhibited by IGF2BP1 reported to date is HIV-1. This review summarizes the current knowledge about the interactions between IGF2BP1 and different virus species. It further recapitulates several findings by presenting analyses from publicly available high-throughput datasets.

## 1. Introduction

Viruses actively reprogram the metabolism of their host cells to support the infection process and facilitate the escape or suppression of host defense mechanisms. To achieve this, the virus relies on interactions with host cell components that are exploited to promote the processes augmenting virus spreading [1]. The reprogramming of cellular metabolism following many viral infections is reminiscent of the metabolic changes observed in tumor development and, consequently, the metabolic reprogramming of certain viruses has been linked to oncogenesis [2]. Virus species described to possess oncogenic potential in humans include Epstein–Barr virus, Kaposi sarcoma-associated herpesvirus, human papillomaviruses, hepatitis B and C viruses, human T lymphotropic virus, and Merkel cell polyomavirus [3,4]. Furthermore, other polyomaviruses, such as simian virus 40 (SV40), as well as adenoviruses have been shown to be capable of transforming animal and human cells [4,5,6]. One important class of host cell components utilized by viruses are RNA-binding proteins (RBPs). These proteins accompany and regulate host as well as viral RNAs through all stages of post-transcriptional gene regulation. Another important class of molecules involved in regulating viral RNAs are microRNAs (miRNAs), small non-coding RNAs that direct the post-transcriptional repression of mRNA targets [7]. Cellular miRNAs can exert anti- or proviral effects and they are often dysregulated upon viral infections [8,9]. For example, miRNA-128 limits the replication and dissemination of the human immunodeficiency virus type 1 (HIV-1), whereas miRNA-34 promotes HIV-1 pathogenesis by inducing the accelerated decay of host proteins influencing the HIV-1 life cycle [10,11]. Virus infections have also been shown to induce post-transcriptional modifications on host genes’ transcripts. N^6^-adenosine methylation (m^6^A) is the most prevalent internal modification found in eukaryotic mRNAs [12]. This modification affects RNA structure and function and was shown to be altered on host genes upon viral infections. On the other hand, viral RNAs are also subject to m^6^A modifications, leading to alterations in virus production and infectivity. However, there seems to be no common regulation pattern and thus, increased methylation can enhance the production of certain viruses, while it reduces the production rates of others [13,14,15,16,17,18].

IGF2BP1, also known as IMP-1, CRD-BP and VICKZ1, is an oncofetal RNA-binding protein regulating tumor and stem cell fate [19,20,21,22,23]. Elevated IGF2BP1 expression has been implicated in the development and progression of various cancers in, e.g., ovary, lung, pancreas, liver and breast and is typically associated with poor prognosis [24,25,26,27,28,29]. IGF2BP1’s main function in cancer appears to be the protection of its mainly pro-oncogenic target RNAs from miRNA-mediated degradation via binding to its four hnRNPK homology (KH) domains [25,30,31,32,33,34,35]. Moreover, it was shown that m^6^A RNA modifications on IGF2BP1 target transcripts result in stronger RNA association and the consequently enforced expression of these pro-oncogenic factors [30,36,37]. By stabilizing the transcription factor E2F1- and E2F-driven transcripts, IGF2BP1 promotes G1/S cell cycle transition [37]. Similarly, distinct tumorigenic viruses drive the expression of S-phase genes by enhancing E2F activity. Papillomavirus E7 proteins, polyomavirus large T antigens and adenovirus E1A proteins are all capable of binding host Rb proteins, leading to a de-repression of E2F transcription factors and, thus, activating genes required for cell cycle progression [5,38,39,40,41].

The first report available on Pubmed regarding an interaction between IGF2BP1 and the components of a virus was published in 2004. Lu et al. reported an association between IGF2BP1 and the RNA genome of the hepatitis C virus [42]. Since then, associations of IGF2BP1 or its vertebrate homologs and several virus species have been published. These include viruses with double-stranded DNA genomes, such as hepatitis B virus and papillomaviruses, single-stranded RNA viruses, such as hepatitis C virus, Zika virus, Ebola virus and the severe acute respiratory syndrome coronavirus 2 (SARS-CoV-2), as well as retroviruses such as HIV-1 and murine leukemia virus (MLV). The respective associations were reported to either impact RNA turnover or translation. In addition, protein–protein interactions leading to the re-localization of viral proteins and altered virus production have been reported. Moreover, virus-mediated epigenetic modifications have been shown to alter RNA-binding activities between IGF2BP1 and its host mRNAs [43,44,45,46,47,48,49].

The oncofetal expression pattern of IGF2BP1 can be recapitulated from various public data sources derived from RNA-sequencing experiments. Healthy tissues express barely detectable amounts of IGF2BP1 mRNA, with only a few exceptions such as testis and kidney (Appendix A). In contrast, the IGF2BP1 mRNA levels are elevated in a multitude of tumors, whereas the inter-sample variation is rather high (Appendix A). Regarding cell types, single-cell RNA-seq data reveal few specialized cell types, mainly embryonic or from adult reproductive tissues as well as distinct cell types found in the kidney, expressing considerable amounts of IGF2BP1 (Appendix A). Thus, for leveraging IGF2BP1, a virus either needs to infect cells expressing sufficient levels of this pro-oncogenic protein or must enforce/induce its synthesis.

This review aimed to summarize the current knowledge about interactions between IGF2BP1 and the components of different virus species and the consequences of these interactions on virus production and infectivity. The study covers IGF2BP1–virus interactions related to the oncogenic hepatitis B/C and human papillomaviruses as well as the non-tumorigenic viruses HIV-1, SARS-CoV-2, Zika virus and Ebola virus and it briefly summarizes reports regarding several non-human pathogenic viruses.

## 2. Materials and Methods

Statistical analyses were performed using R and images were created using the R-package ggplot2 [50,51], if not stated otherwise. Network drawings were created using Cytoscape [52]. Bulk RNA-seq data from the TCGA project were downloaded as files containing FPKM values via the GDC data portal (https://portal.gdc.cancer.gov/repository). The infection status of individual patient samples was extracted from supplementary tables of the accompanying reports. HBV/HCV: TCGA cohort LIHC, column “Hepatitis B”/“Hepatitis C” [53]; HPV: TCGA cohort CESC, column “CLIN:HPV_status” [54]. Liquid association coefficients were calculated using the R-package LiquidAssociation [55]. Survival analyses were performed as logrank tests implemented in the R-package survival [56] using the TCGA RNA-seq expression data of primary tumor samples. The CCLE bulk RNA-seq data of cell lines were obtained from DepMap, downloaded via the R-package ExperimentHub [57,58,59]. GTEx bulk RNA-seq data were obtained by downloading GTEx v8 TPM level data from the GTEx portal (https://www.gtexportal.org/home/) [60].

For the analysis of bulk-RNA-seq data from individual projects, fastq files were downloaded from NCBI GEO (https://www.ncbi.nlm.nih.gov/geo/). The quality of the fastq files was assessed using FastQC (https://www.bioinformatics.babraham.ac.uk/projects/fastqc/). If considerable amounts of remaining sequencing adapters or low-quality read ends were detected, these were clipped off using Cutadapt [61]. Sequencing reads were aligned to the human genome (UCSC hg38) using HiSat2 [62]. Alignments in the obtained bam files were sorted, indexed and secondary alignments were filtered out using samtools [63]. FeatureCounts [64] was used for summarizing the gene-mapped reads. Ensembl (GRCh38.100 [65] was used as the annotation basis. Differential gene expression was determined using the R package edgeR utilizing the trimmed mean of M-values [66,67] normalization. A false discovery rate (FDR)-adjusted *p*-value below 0.05 was considered the threshold for the determination of differential gene expression. Sequencing reads not mapped to the human reference genome were subsequently mapped against the respective virus genes/genomes using Hisat2. The following RefSeq sequences were used: HCV—NC_004102.1; HPV16 E7—NC_001526:7604-7900; HPV18 E7—NC_001357.1:590-907; SARS-CoV-2—NC_045512.2. Normalized expression values were obtained as fraction of the reads mapped to the respective gene/genome of the library size multiplied by 1 × 10^6^.

Single-cell RNA-seq data of CD4^+^ T cells were downloaded as R-object files along with cell cluster information from PanglaoDB (https://panglaodb.se/). The R-package Seurat V4 [68] was used for subsequent processing steps, including cell filtering (the minimum number of expressed genes/cell: 100; maximum number of expressed genes/cell: 2000; maximum percentage of mitochondrial gene expression/cell: 10) and count normalization using the “LogNormalize” method and scaling factor 10,000.

## 3. Human Oncogenic Viruses

### 3.1. Hepatitis B Virus

Hepatitis B virus (HBV) is an enveloped DNA virus mainly infecting hepatocytes. HBV infection can result in the integration of the viral genome into the host genome and chronic infection is the main cause of liver cirrhosis and hepatocellular carcinoma (HCC) worldwide [69,70,71]. HBV’s carcinogenic potential is thought to be facilitated by different mechanisms such as mutagenesis due to the integration of viral DNA into host cancer genes, the promotion of genomic instability either via the integration of viral DNA or via the activity of certain viral proteins and due to the interference of normal cellular functions by viral proteins [72]. The regulatory HBx protein promotes HBV transcription as well as replication and contributes to the transformation of hepatocytes via multiple mechanisms. HBx directly interacts with a multitude of cellular proteins involved in proliferation, cell death, transcription and DNA repair [72]. One aspect of the HBx-mediated promotion of tumor progression is via the activation of the proto-oncogenic transcription factor c-Myc. While Li et al. described the induction of c-Myc expression via HBx by the activation of Ras/Raf/ERK1/2 cascades, Yan et al. reported that the HBx protein promotes tumor growth by facilitating c-Myc RNA stabilization via IGF2BP1 [49,73]. The latter described an interaction between HBx and the DNA methyltransferase DNMT3A leading to a hypermethylation of the promoter of the protein tyrosine phosphatase PTPN13, which, in turn, leads to a reduced PTPN13 transcription. In combination with the finding that PTPN13 directly interacts with the IGF2BP1 protein and that PTPN13 overexpression impairs the binding between IGF2BP1 and the c-Myc transcripts, Yan et al. concluded that PTPN13 protein competes with c-Myc RNA for IGF2BP1 association resulting in elevated c-Myc mRNA decay. Thus, the epigenetic silencing of PTPN13 mediated by HBV’s HBx protein enhances HCC proliferation by increasing c-MYC mRNA stability in an IGF2BP1-dependent manner [49]. In addition, Yan et al. described a dose-independent upregulation of IGF2BP1 RNA and protein by HBx [49], whereas Wang et al. observed a downregulation of let-7 miRNA family members by HBx [74]. Moreover, You et al. discovered that the LIN28B promoter was activated by HBx via c-Myc [75]. Intriguingly, IGF2BP1 has been shown to bind and thus stabilize LIN28B RNA and both, this stabilization and the activation of LIN28B via c-Myc were shown to repress let-7 miRNAs [31,76]. Altogether, this points to a regulatory network in which HBx, c-Myc, IGF2BP1 and LIN28B act synergistically to suppress let-7 action or biogenesis. Thus, the expression of all these factors is promoted by viral infection and fosters cell proliferation (Figure 1A).

The downregulation of PTPN13 in HCC upon HBV infection could be recapitulated from TCGA HCC RNA-seq data [53]. However, the difference in RNA expression levels between patients with and without hepatitis B infection was rather moderate; however, there was an obviously lower PTPN13 expression in primary tumors compared to normal tissues. C-Myc expression was higher in HBV-positive HCC samples compared to non-infected samples; however, generally elevated in normal tissue samples. IGF2BP1 expression was significantly elevated in tumor samples and in both, tumor and non-tumor tissues, IGF2BP1 abundance appeared to be increased by HBV infection (Figure 1B).

Liquid association (LA), a technique to assess the degree of interference in an expression correlation between two genes by a third gene [77], indeed revealed a negative influence of the PTPN13 expression on the correlation between IGF2BP1 and c-Myc (negative LA coefficient) in HCC-derived, HBV-positive cell lines. This was further supported by a very strong Pearson correlation between IGF2BP1 and c-Myc in the cell lines showing a low PTPN13 RNA expression compared to cell lines with higher PTPN13 expression (cell lines separated by median PTPN13 RNA expression; Figure 1C). However, although negative LA coefficients were also observed in HBV-positive HCC-patient samples, the non-tumor tissues of HBV-infected patients as well as tumor samples from non-infected patients showed a positive LA coefficient, suggesting that the interference of PTPN13 on the binding between the IGF2BP1 protein and c-Myc RNA is not only dependent on the expression levels of PTPN13, but also involves additional, difficult to identify, regulatory constraints (Figure 1D). The hypothesis that PTPN13 binding to IGF2BP1 competes with c-Myc binding leads to the tempting speculation that the binding of PTPN13 to IGF2BP1 reduces the interaction between IGF2BP1 and other target transcripts as well. To test this hypothesis, the LA coefficients of IGF2BP1 and a set of 117 published target candidates in dependence of PTPN13 were calculated. The criteria for selecting genes as target candidates comprised a consistent downregulation upon IGF2BP1 knockdown in several cancer-derived cell lines, a positive expression correlation with IGF2BP1 in tumors as well as reported IGF2BP1 binding sites [35]. Positive as well as negative LA coefficients were obtained for distinct target candidates; however, in general, LA coefficients were lowest in the HBV positive tumor samples (Figure 1E). Thus, PTPN13 might indeed interfere with the binding of IGF2BP1 and some of its targets; however, further investigations are required. Patient survival in HCC tends to be negatively associated with IGF2BP1 levels, but an additional influence of HBV on survival probabilities could not be conclusively assessed from TCGA data. The HBV-positive cohort was rather small (n = 20), the logrank test derived *p*-value was not significant and the Kaplan–Meier curves crossed, thus, allowing only rather unassertive conclusions (Appendix A).

**Figure 1 viruses-15-01431-f001:**
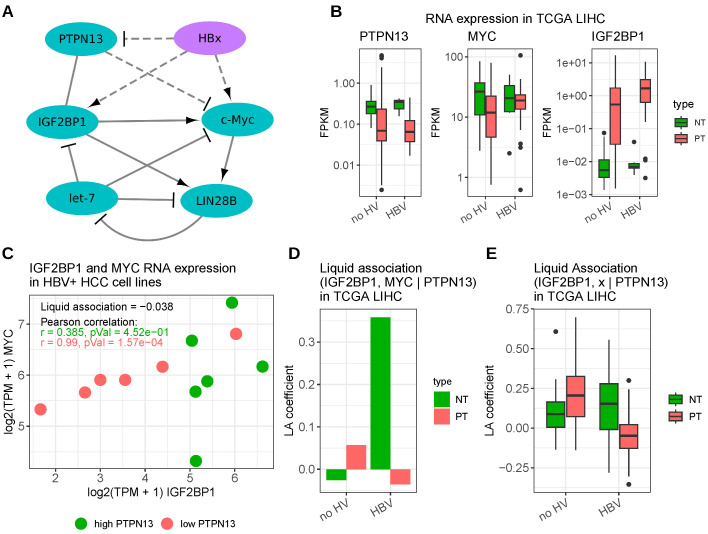
Interaction between HBV and IGF2BP1. (**A**) Scheme depicting the published interactions between HBV’s HBx protein and host cell components, cf. [31,49,74,75,76]. Solid line—direct interaction; dashed line—indirect interaction. (**B**) Distribution of RNA expression values of PTPN13, c-Myc and IGF2BP1 derived from HCC-RNA-seq samples (TCGA, cohort LIHC). (**C**) IGF2BP1 and c-Myc RNA expression in RNA-seq samples of HCC cell lines (data derived from CCLE data accessed via DepMap [57,58]; HBV-positivity was assessed from Expasy Cellosaurus [78]). (**D**) Liquid association (LA) coefficients of the correlation between IGF2BP1 and c-Myc in dependence of PTPN13 in TCGA HCC samples. (**E**) LA coefficient distributions of the correlation between IGF2BP1 and 117 published target candidates in dependence of PTPN13 in TCGA HCC samples. *no HV*—samples without detected hepatitis virus infection; *HBV*—samples with exclusively detected hepatitis B infection; *NT*—normal tissue; and *PT*—primary tumor tissue.

### 3.2. Hepatitis C Virus

The hepatitis C virus (HCV) is the main causative agent of non-A and non-B hepatitis [45] and besides HBV, constitutes another major risk factor for the development of liver cirrhosis and hepatocellular carcinoma [71,79,80]. The primary target cells of the virus are hepatocytes [81].

The positive-strand RNA genome of HVC contains an internal ribosomal entry site (IRES) in its 5’ untranslated region (5’UTR), facilitating cap-independent translation via the direct recruitment of the 40S ribosomal subunit and the eukaryotic initiation factor 3 (eIF3) [82,83].

In 2004, Lu et al. identified IGF2BP1 (CRDBP) as well as its paralogs IGF2BP2 and IGF2BP3 as HCV IRES-interacting proteins [42]. The binding of IGF2BP1 to the HCV-IRES was confirmed by Weinlich et al. in 2009. Furthermore, these authors found that IGF2BP1 can also bind to the HCV 3′UTR in vitro and that IGF2BP1 knockdown in HCC-derived Huh-7 cells leads to strongly decreased HCV IRES-mediated translation rates that were even further reduced in the absence of the HCV 3′UTR. However, the stability of the used reporter mRNAs was not reduced upon IGF2BP1 knockdown. Consistently, the addition of IGF2BP1 to translation competent extracts of primary rat hepatocytes increased the translation rates of an HCV reporter mRNA containing the HCV 5′ and 3′UTRs. In addition, a more than two-fold increase in the IGF2BP1 protein level was observed in Huh-7 cells, expressing an HCV replicon containing the nonstructural proteins NS3, NS4A, NS4B, NS5A and NS5B [45]. Moreover, Cheng et al. reported the anti-HCV activity of the let-7 miRNA family, as well as the negative regulation of IGF2BP1 expression by these miRNAs [84], providing further support for a regulatory role of an IGF2BP1-let-7 antagonism [31,85]. Figure 2A depicts the relationships between the HCV genome, IGF2BP1 and let-7 miRNAs. Another aspect of IGF2BP1 binding to the HCV RNA was reported by Li et al. They reported a competition in binding to the 5′ end of the HCV RNA between IGF2BP1 and miR-122, a liver-specific miRNA reported to promote HCV replication and translation [86,87,88,89]. After siRNA-mediated IGF2BP1-knockdown in Huh7.5 cells expressing HCV genomic RNA, significant reductions were observed in both HCV replication as well as HCC cell proliferation [86]. Bradrick et al. confirmed the binding of IGF2BP1 in the region of the miR-122 binding sites and, furthermore, found that the binding of IGF2BP1 was compromised by the deletion of the miR-122 binding sites; however, point mutations in these sites had no apparent effect on IGF2BP1 binding, suggesting the miR-122 independent binding of IGF2BP1 [89].

While there was no obvious difference in IGF2BP1 RNA expression in the TCGA HCC samples with and without detected HCV infection (Appendix A), a positive correlation between IGF2BP1 RNA expression and the RNA levels of HCV can be seen in publicly available RNA-seq samples of HCV-infected Huh7.5.1 cells (Figure 2B, the data for which were obtained from [90]). Moreover, IGF2BP1 expression levels reached a moderate yet significant elevation after 8 days of infection in these cells (Figure 2C). Although not significant, survival analyses in the TCGA HCC cohort indicated a strong negative impact of IGF2BP1 expression on patients’ overall survival (Appendix A).

### 3.3. Human Papillomavirus

Human papillomaviruses (HPVs) are a diverse set of more than 200 types of dsDNA viruses. They comprise five evolutionary groups (Alpha, Beta, Gamma, Mu, Nu) of which the Alpha group contains members causing rather harmless warts, but also the so-called high-risk (HR-) HPVs that are associated with several cancer types, including cervical cancer, several other anogenital cancers, as well as head and neck, lung and breast cancer [91,92,93,94]. Almost 100% of cervical cancers are attributable to HR-HPVs with HPV16 and 18 accounting for 71% of cases worldwide [95] and in total are estimated to cause more than 5% of all human cancers [96]. Nonetheless, cancer progression is a rather rare event following persistent HPV infection [97]. Notably, several studies have shown that HPV infection in head and neck cancers is associated with a better prognosis [98,99,100,101]. Cancer progression upon HR-HPV infection is caused by the increased expression of the viral oncoproteins E6 and E7 that activate the cell cycle, inhibit apoptosis, and allow for the accumulation of DNA damage [97]. HPV16 E7 can bind members of the well-known tumor suppressor and cell cycle regulatory retinoblastoma protein family (Rb), leading to the degradation of these proteins [38,39,40]. The binding/degradation of Rb releases and activates E2F transcription factors that drive the expression of S-phase genes [102]. Furthermore, E7 can directly bind and activate E2F1 in an Rb-independent manner [103]. HPV E6 protein complements the actions of E7 by targeting p53 for degradation and thus, preventing cell growth inhibition [102].

Similar to HPVs’ E7, IGF2BP1 supports E2F1 activity, albeit via different mechanisms. Müller et al. could show that IGF2BP1 promotes G1/S cell cycle transition by stabilizing mRNAs encoding positive regulators of this checkpoint including E2F1 and E2F-driven transcripts [37]. Wang et al. reported the m^6^A-dependent binding of HPV16 E7 mRNA by IGF2BP1, leading to a stabilization and, thus, increased the expression of this RNA in HPV16 positive cell lines. Moreover, this E7-IGF2BP1 complex was shown to be vulnerable to heat stress, since exposing cells to moderate heat led to a decrease in E7 as well as IGF2BP1 and to an upregulation of Rb, p53 and p21 proteins. In consequence, heat treatment was shown to reduce the proliferation and migration in HPV16-positive cells and reduced tumor growth in cell line-based xenograft mouse models [43]. Figure 3A depicts the described relations between E7 and IGF2BP1. Interestingly, besides the downregulation of p53 and Rb in HPV-positive lung cancer tissues, Hussen et al. observed a downregulation of PTPN13 and let-7 miRNAs [104]. Thus, it is tempting to speculate that regulatory circuits involving PTPN13, let-7 miRNAs, c-Myc and IGF2BP1 are also relevant in the HPV-associated malignancies (cf. Section 3.1). Regarding patient prognosis, Laban et al. observed a significant association between IGF2BP1 (IMP-1) antibody response and the shorter overall survival of patients with HPV-positive head and neck squamous cell carcinomas (HNSCC), which was not seen in HPV-negative patients [105].

A strong positive correlation between HPV E7 and IGF2BP1 RNA could be recapitulated from RNA-seq data in a panel (n = 10) of different cervical squamous carcinoma-derived cell lines expressing either HPV16 (CaSki, n = 4; SiHa, n = 3) or HPV18 (HeLa, n = 3) proteins (data derived from [106], Figure 3B). Furthermore, a slightly elevated, yet not significant, expression of IGF2BP1 RNA was observed in HPV-positive cervical cancer samples deposited by the TCGA (median FPKM = 0.08), and compared to HPV-negative samples (median FPKM = 0.03, Figure 3C). Considering the complete cervical cancer cohort, IGF2BP1 failed to yield conclusive results in terms of survival probability. However, HPV status remained unknown for many of these samples (Appendix A). Considering only HPV-positive samples indicated a strong, yet not significant trend that IGF2BP1 is associated with reduced survival, whereas studies in the few HPV-negative samples remained non-conclusive (Appendix A).

## 4. Human Non-Oncogenic Viruses

### 4.1. Human Immunodeficiency Virus Type 1

The human immunodeficiency virus type 1 (HIV-1) is the causative agent of the acquired immunodeficiency syndrome (AIDS). HIV-1 can infect CD4^+^ T cells, macrophages and dendritic cells [107]. The major structural component of HIV virions is the Gag protein. This protein is essential for virus assembly and release. It is further involved in membrane binding, the incorporation of the viral genome and Env proteins into the virions as well as in budding and release [108].

Zhou et al. discovered an association between IGF2BP1 and the HIV-1 Gag protein. The binding between the two proteins was shown to be primarily facilitated by IGF2BP1’s KH3 and KH4 domains and a zinc finger motif in the Gag nucleocapsid (NC) domain [47]. KH domain-dependent IGF2BP1-Gag binding was confirmed by Milev et al. [109]. Furthermore, IGF2BP1 expression was shown to inhibit the production of virus particles; however, effective inhibition relied on the presence of all IGF2BP1 domains. Moreover, Zhou et al. observed the incorporation of IGF2BP1 into virus particles and that the infectivity of virions produced in IGF2BP1 overexpressing cells was reduced in a primarily KH domain-dependent manner [47]. In a following work, Zhou et al. found that IGF2BP1 also interacts with HIV-1’s regulatory Rev protein [48]. The Rev protein binds and mediates the cytoplasmic transport of late-phase partially spliced viral RNAs as well as the unspliced whole RNA genome that is packaged into the budding virions [110]. The association between Rev and IGF2BP1 resulted in a re-localization of a substantial amount of Rev from nucleoli to the cytoplasm where it co-localized with IGF2BP1. This re-localization resulted in a disturbed ratio of the (multiply) spliced and unspliced viral RNAs found in the cytoplasm [48]. In addition, Zhou et al. infected a CD4^+^ T cell line (SupT1), ectopically expressing FLAG-IGF2BP1, with wild-type HIV-1, and observed a reduced production of the virus. Interestingly, although the authors could confirm the inhibitory role of IGF2BP1 on HIV-1 virus production, they also found that IGF2BP2 showed an even stronger effect on HIV-1 inhibition, whereas IGF2BP3 did not exert an effect on infectivity [48]. For the interaction studies, Gag and Rev were transfected into non-hematopoietic cells (HEK293T and HeLa) and the T cell line used for investigating the effect of IGF2BP1 on virus production was engineered to ectopically produce the protein, raising the question about the physiological relevance of the interactions between IGF2BP1 and HIV-1.

Single-cell RNA-seq data of primary CD4^+^ T cells either uninfected or infected with HIV-1 indicate barely any expression of IGF2BP1 and its paralogs (Figure 4A). Bulk RNA-seq data of blood cells, obtained from the GTEx project [60], accordingly, show a very low RNA expression of IGF2BP1. Furthermore, an upregulation of IGF2BP1 in leukemia, as frequently observed in other cancer types, cannot be seen in the TCGA acute myeloid leukemia cohort RNA-seq data. However, although the IGF2BP2 expression was also barely detected in single T cells, a moderate to high expression of IGF2BP2 mRNA was observed in the blood bulk RNA-seq and leukemia dataset, suggesting that the reported inhibitory effect of the IGF2BP proteins on HIV in T cells might be more relevant for IGF2BP2 (Figure 4B,C). According to single-cell RNA-seq data provided by the human protein atlas [111], a macrophage population found in the kidney shows moderate IGF2BP1 expression. In addition, this macrophage subpopulation was also positive for CD4, CXCR4 as well as CCR5, i.e., the receptors utilized by HIV-1 to enter the cells. However, these macrophages also express IGF2BP2 at even higher levels (Appendix A). Thus, such kidney-residing macrophages might indeed be susceptible for an interplay between IGF2BP1/2 and HIV-1; however, this needs further investigation.

### 4.2. Severe Acute Respiratory Syndrome Coronavirus 2

The severe acute respiratory syndrome coronavirus 2 (SARS-CoV-2) is a single-stranded RNA virus, causing the coronavirus disease 19 (COVID-19) that became a worldwide pandemic in 2020. In most cases, SARS-CoV-2 infection causes mild cold symptoms; however, a significant proportion of patients develop severe symptoms, including acute respiratory distress (ARDS), pneumonia, renal failure, cardiac complications and even multi-organ failure. Furthermore, reports about persistent symptoms that may last for several months following an acute SARS-CoV-2 infection, designated as post-COVID syndrome, are accumulating [113,114,115]. The structural spike (S) protein of SARS-CoV-2 is equipped with a receptor-binding domain (RBD) mediating direct contact with the angiotensin-converting enzyme 2 (ACE2) receptor of susceptible host cells. However, for viral entry, the spike protein has to be primed by host cell proteases TMPRSS2 or cathepsin B/L [116]. Since SARS-CoV-2 enters the body via the respiratory tract, airway and alveolar epithelial cells as well as vascular endothelial cells and alveolar macrophages belong to the first targets of viral entry. However, cells from multiple extra-pulmonary tissues also express ACE2 and TMPRSS2 and thus represent possible target cells for SARS-CoV-2 [114,117,118].

Zhang et al. observed a stimulation of IGF2BP1 RNA expression upon SARS-CoV-2 infection in cell lines and patient lung samples. In addition, they found that the knockdown as well as knockout of IGF2BP1 in different cell lines led to reduced levels of viral RNA in these cells after infection with SARS-CoV-2. Importantly, IGF2BP1 was shown to bind and stabilize a subgenomic RNA encoding the SARS-CoV-2 S protein in vitro and thus, promoting S protein translation [44].

The inspection of two independent publicly available bulk RNA-seq datasets of SARS-CoV-2 infected colon adenocarcinoma derived Caco-2 and primary human bronchial epithelial cells (HBECs), respectively, led to inconclusive results. Considerable amounts of SARS-CoV-2 genomic RNA were detectable upon the infection in both cell lines; however, the observed amounts varied tremendously. Around 30% of all sequencing reads obtained from the Caco-2 cells were mapped to the SARS-CoV-2 genome at 7, 12 and 24 h after infection. In contrast, less than 1% of sequencing reads could be mapped to the virus genome in infected HBEC samples. IGF2BP1 RNA levels also varied by about three orders of magnitude between the two cell lines. The normalized expression values of IGF2BP1 transcripts and SARS-CoV-2 genomic RNA exhibited negative, yet not significant, Pearson correlation in infected cells from both cell lines (Figure 5A,D). Nonetheless, IGF2BP1 RNA levels tended to increase over time in infected cells, whereas the levels of SARS-CoV-2 genomic RNA started to decrease in Caco-2 cells between 12 and 24 h after infection, while they continued to increase in HBECs until the final measurements (96 h, Figure 5B,E). Compared to mock-infected (Caco-2)/uninfected (HBECs) cells, IGF2BP1 levels started to increase in both cell lines and reached almost four-fold higher levels (log_2_ fold change = 1.89) 48 h after infection in the HBECs. However, this increase was not significant (FDR adjusted *p*-value = 1), due to high variation in the expression in the biological replicates. Despite IGF2BP1 levels continuing to rise in infected cells, the rise in the expression of IGF2BP1 compared to uninfected HBEC cells was attenuated, since IGF2BP1 levels also increased in the uninfected cells (Figure 5C,F). However, IGF2BP1 RNA expression levels in the HBECs in total were rather low and, thus, detected differences might be unreliable and the consequence of data noise. Considering cell-type-specific RNA expression obtained from human protein atlas single-cell RNA-seq data, human kidney cells displayed salient co-expression patterns. Distinct sub-populations of proximal tubular cells express at least moderate levels of IGF2BP1 as well as ACE2, TMPRSS2 and cathepsin-B (CTSB) RNA (Appendix A). Thus, due to a basal expression of IGF2BP1 in healthy human kidneys, they could potentially be organs vulnerable for infection by SARS-CoV-2 that might be intensified by IGF2BP1. However, further investigations are required. Moreover, it needs to be addressed if SARS-CoV-2 infections could lead to IGF2BP1 de novo synthesis, which might explain the observed elevated levels in patient lung samples.

### 4.3. Other Human Viruses

Besides SARS-CoV-2, Zhang et al. observed binding between IGF2BP1 and two other RNA viruses, namely Zika virus (ZIKV) and Ebola virus (EBOV) [44]. Similarly to SARS-CoV-2, the authors found a significant reduction in viral ZIKV RNA levels in infected cells upon IGF2BP1 knockdown and knockout. However, in contrast to SARS-CoV-2, they did not observe significant differences in the degradation rates of the ZIKV genomic RNA upon IGF2BP1 knockout; thus, they concluded that there was no stabilization effect, despite the fact that they observed that IGF2BP1 also promoted the translation of this virus [44]. The binding of IGF2BP1 to EBOV RNA was shown by Zhang et al. and Fang et al. However, while Zhang et al. did not observe an impact of IGF2BP1 knockdown on EBOV RNA levels, Fang et al. reported a significant reduction in infection rates after knocking down IGF2BP1 independently with two out of three used siRNAs [44,120].

## 5. Non-Human Viruses

Besides the above-reported human-pathogenic viruses, IGF2BP1 homologs were reported to promote the viral spread of several viruses infecting other animals.

Chen et al. reported the direct binding of IGF2BP1 to the 3′UTR of Duck hepatitis A virus type 1 (DHV-1/DHAV), a highly fatal, rapidly spreading virus infecting young ducklings [121], which led to the increased IRES-mediated translation efficiency of viral proteins in a duck embryo epithelial cell line. Furthermore, they reported that the presence of DHV-1 proteins in the cytoplasm of the duck embryo epithelial cell line led to increased IGF2BP1 protein levels [46]. The nonstructural p17 protein of the avian reovirus (ARV), another important poultry pathogen, was shown to interact with IGF2BP1 and the over-expression of IGF2BP1 in chicken cells was associated with an increase in viral replication. Furthermore, the mRNA levels of IGF2BP1 were increased upon ARV infection in chicken cells [122]. Another poultry-infecting virus with a reported connection to IGF2BP1 is the Marek’s disease virus (MDV), a highly contagious herpesvirus capable of inducing T-cell lymphomas in chicken [123]. IGF2BP1 expression was found to be increased in a chicken inbred line, highly susceptible to MDV, whereas, in a MDV-resistant cell line, IGF2BP1 RNA expression was hardly detectable. The difference in RNA expression was concordant with the increased activating histone methylation marks (H3K4me3) around the IGF2BP1 transcription start site found in the MDV susceptible chicken samples. Upon MDV infection, IGF2BP1 expression was significantly reduced in thymus cells of these animals [124]. However, it remained open, if the elevated IGF2BP1 expression was connected to higher susceptibility to the virus. Mai and Gao observed an IGF2BP1-dependent increase in the production of infectious murine leukemia virus (MLV) vectors, a retroviral vector system widely used for gene transfer. By binding and stabilizing the viral genomic RNA, IGF2BP1 increased its incorporation into virions. However, for the wild-type virus, only a modest effect was measured, since, although still increasing the incorporation of genomic RNA, IGF2BP1 showed an inhibitory effect on the production of viral proteins [125]. Similarly to HIV-1, IGF2BP1 protein was found to also be incorporated into the virion particles of this retrovirus [47,125]. Jefferson et al. identified IGF2BP1 as a binding partner of the N-terminal protease N^pro^ of the classical swine fever virus (CSFV)/Pestivirus C by the GST pulldown assay and subsequent mass spectrometry. Since the number of detected peptides was increased in the presence of zinc, the authors concluded that the binding involved the interaction with the TRASH motif contained in the interaction domain of N^pro^ [126].

## 6. Discussion

The RNA-binding protein IGF2BP1 has been reported to interact with RNA and proteins from a variety of viruses from different families, such as Hepadnaviridae (HBV), Flaviviridae (HCV, ZIKV, CSFV), Papillomaviridae (HPV), Retroviridae (HIV-1, MLV), Coronaviridae (SARS-CoV-2), Filoviridae (EBOV), Picornaviridae (DHV-1), Reoviridae (ARV) and Herpesviridae (MDV). Associations with IGF2BP1 include the binding of viral transcripts leading to their stabilization (HPV, SARS-CoV2, MLV; [43,44,125]), but also stability-independent enhanced translation (HCV, DHV-1; [45,46]). Furthermore, the binding of IGF2BP1 to viral proteins led to alterations in virus production, infectivity, or replication (HIV-1, ARV; [47,122]). Binding to IGF2BP1 usually results in proviral effects [43,44,45,46,49,86,120,122,126]. Only the binding to proteins of HIV-1 was reported to reduce virus production and infectivity [47,48]. Besides the putative influence on the natural infection process, an inhibitory effect of IGF2BP1 on HIV-1 production, facilitated by binding to Gag and Rev proteins, might also impact research and gene therapy approaches using lentiviral vectors comprising HIV-derived genes. Regarding the extent of reported virus families, the diversity of utilized molecular mechanisms and IGF2BP1’s function in promoting cell proliferation, it seems likely that even more virus species benefit from elevated IGF2BP1 expression. However, since IGF2BP1 is an oncofetal protein [19,20,21,22,23], questions about the physiological relevance of the reported interactions between IGF2BP1 and non-tumor viruses, primarily investigated in tumor-derived or embryonic cell lines, needs to be addressed by further analyses. Remarkably, however, upon infection with HBV, HCV, SARS-CoV-2, DHV-1 and ARV, increasing levels of IGF2BP1 RNA and/or protein have been observed [44,45,46,49,122]. This leads to the unsettling conclusion that infection with these, in part so far, not described as tumor-promoting viruses, might drive the expression of a protein that is known to be involved in the development and progression of multiple types of tumors [24,25,26,27,28,29]. The underlying mechanisms leading to the enhanced IGF2BP1 expression were not investigated and, thus, still need to be elaborated. On the other hand, as for cancer, the oncofetal expression pattern of IGF2BP1 could be exploited for developing novel anti-viral therapies, since the depletion/inactivation of this protein in adult tissues should have only little effects on normal cellular physiology. For example, the opposing functions of IGF2BP1 and miRNAs of the let-7 family on the spreading of viruses such as HBV, HCV and HPV might be utilized, e.g., by enhancing let-7 expression. Interferon alpha and Interleukin-28B treatment have been shown to increase cellular let-7 levels, decrease the IGF2BP1 expression and exert an anti-HCV effect [84]. Furthermore, small molecule inhibitors of IGF2BP1 function, including BTYNB or Cucurbitacin B, that have recently shown promising anti-tumor effects [37,127,128], should be tested for their impact on the production and infectivity of the virus species reported to benefit from IGF2BP1 expression.

In conclusion, IGF2BP1 can be recruited by a variety of viruses from distinct families to promote their spreading, and thus, anti-tumor strategies aiming to inhibit the expression or function of IGF2BP1 should be tested for anti-viral effects as well. Table 1 briefly summarizes the effects that IGF2BP1 exerts on the components of the described viruses as well as the impact of these viruses on IGF2BP1.

## 7. Future Directions

Despite only few studies dedicated to IGF2BP1–virus cross-talk, the broad range of virus families covered by these reports suggests a widespread role of IGF2BP1 in viral infection and the impact of viruses on IGF2BPs, especially in cancer. Recently developed data repositories, such as NCBI’s sequence read archive (SRA, https://www.ncbi.nlm.nih.gov/sra) or the European nucleotide archive (ENA, https://www.ebi.ac.uk/ena/), provide tremendous amounts of publicly available high-throughput sequencing data that could be a promising starting point for the discovery of new IGF2BP–virus interactions. The examination of appropriate RNA-seq datasets, provided by these repositories, could deliver new insights about the presence of viruses, e.g., in cancerous tissues but also about changes in IGF2BP expression upon virus infection. Correlations between IGF2BP1 or its paralogs and viral RNAs, derived from these studies, can be used to hypothesize about putative direct interactions and indirect cross-talk impacting the viral life cycle. Furthermore, CLIP-seq (cross-linking and immunoprecipitation) datasets could provide valuable information about viral RNAs bound to IGF2BPs. Additionally, binding motifs derived from these datasets can be used to scan viral genomes for the occurrences of putative IGF2BP binding sites. Indirect interactions such as the downregulation of PTPN13 by viral proteins leading to an IGF2BP1-dependent upregulation of c-Myc should also be further exploited. While investigated in detail for HBV [49], Hussen et al. described a reduction in PTPN13 levels upon HPV infection; however, they did not provide a mechanistic explanation [104]. Moreover, we observed a downregulation of PTPN13 as well as a negative influence on the association between IGF2BP1 and c-Myc in HCV-infected TCGA HCC samples—both effects were stronger pronounced than in HBV-positive samples (Appendix A). Thus, further investigations regarding interactions between viruses, IGF2BP1, PTPN13 and c-Myc levels are pending. Respective studies could reveal potential therapeutic options, especially in HCV/HPV-associated cancers. Recent reports provide strong evidence that RBP–RNA interactions are per se druggable and disrupting these interactions may provide benefits for viral infection-associated cancerous diseases [129].

## Figures and Tables

**Figure 2 viruses-15-01431-f002:**
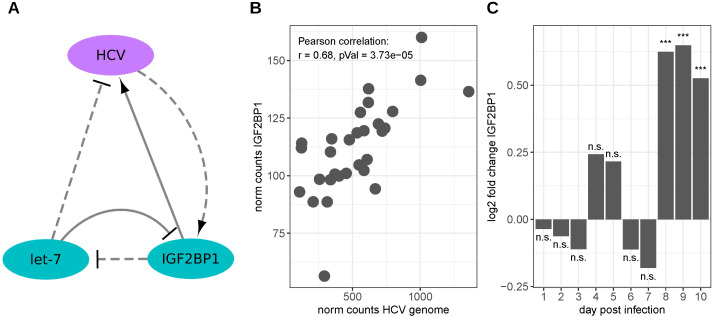
Interaction between HCV and IGF2BP1. (**A**) Scheme depicting the published interactions between the HCV genomic RNA and the host cell components, cf. [31,42,45,84,85]. Solid line—direct interaction, dashed line—indirect interaction. (**B**) Expression levels of IGF2BP1 and HCV RNA in Huh7.5.1 cells infected with HCV (data derived from [90]). (**C**) IGF2BP1 fold changes in HCV-infected Huh7.5.1 cells compared to uninfected cells (data derived from [90]). n.s.: FDR adjusted *p*-value ≥ 0.05, ***: FDR adjusted *p*-value < 0.001.

**Figure 3 viruses-15-01431-f003:**
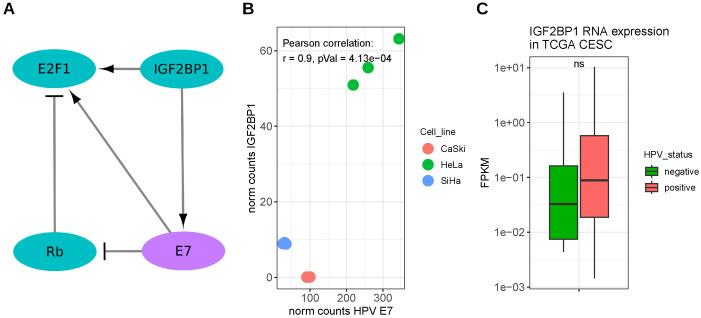
Interaction between HPV and IGF2BP1. (**A**) Scheme depicting published interactions between the HCV’s E7 protein and host cell components, cf. [37,38,39,40,43,102,103]. Solid line—direct interaction. (**B**) RNA expression levels of IGF2BP1 and HPV E7 in HPV-positive cervical cancer cell lines (data derived from [106]). (**C**) IGF2BP1 expression values in cervical cancer samples (TCGA cohort CESC [54]). ns.: Wilcox-test *p*-value ≥ 0.05.

**Figure 4 viruses-15-01431-f004:**
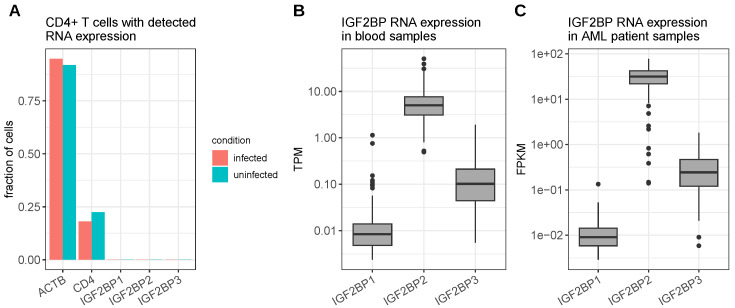
IGF2BP RNA expression in blood cells and leukemia. (**A**) Fraction of CD4^+^ T cells uninfected or infected with HIV-1 with detectable RNA expression (normalized read counts > 0) of the indicated proteins (data derived from single-cell RNA-seq; SRA accession: SRP134859). (**B**) RNA expression levels of the human IGF2BP proteins in bulk RNA-seq data of blood cells (data derived from GTEx [60]). (**C**) RNA expression levels of the human IGF2BP proteins in the bulk RNA-seq data of peripheral blood cells from acute myeloid leukemia patients (data derived from the TCGA LAML cohort [112]).

**Figure 5 viruses-15-01431-f005:**
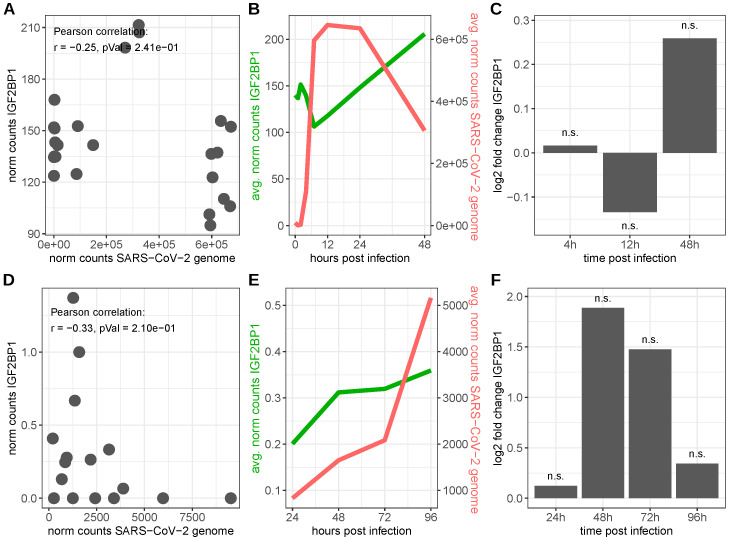
IGF2BP1 RNA expression in SARS-CoV-2-infected cells. Upper row: SARS-CoV-2-infected Caco-2 cells (data derived from GEO series GSE217504), lower row: SARS-CoV-2-infected primary human bronchial epithelial cells (HBECs, data derived from [119]). (**A**,**D**) RNA expression levels of IGF2BP1 and the SARS-CoV-2 genome in SARS-CoV-2-infected cells. (**B**,**E**) Time-point-averaged RNA expression levels of IGF2BP1 and the SARS-CoV-2 genome at the indicated time after infection. (**C**,**F**) IGF2BP1 fold changes in SARS-CoV-2 infected cells compared to mock-infected (Caco-2)/uninfected (HBECs) cells. n.s.: FDR adjusted *p*-value ≥ 0.05.

**Table 1 viruses-15-01431-t001:** Effects of IGF2BP1–virus interactions.

Virus	Impact of IGF2BP1 on Virus	Impact of Virus on IGF2BP1
HBV		Binding to c-Myc mRNA ↑ [49]
		RNA/protein expression ↑ [49]
HCV	Translation ↑ [45]	Protein expression ↑ [45]
	Replication ↑ [86]	
HPV	Stability of E7 RNA ↑ [43]	Heat stability ↓ [43]
HIV-1	Virus production↓ [47,48]	
	Infectivity ↓ [47,48]	
SARS-CoV-2	RNA levels ↑ [44]	RNA expression ↑ [44]
	Stability of S RNA ↑ [44]	
ZIKV	RNA levels ↑ [44]	
EBOV	Infectivity ↑ [120]	
DHV-1	Translation ↑ [46]	Protein expression ↑ [46]
ARV	Replication ↑ [122]	RNA expression ↑ [122]
MDV		RNA expression ↓ [124]
MLV	stability of genomic RNA ↑ [125]	
	Protein levels ↓ [125]	

## Data Availability

Bulk-RNA-seq data from individual projects were downloaded from NCBI GEO (https://www.ncbi.nlm.nih.gov/geo/). HCV: GSE126831, HPV: GSE158033 SARS-CoV-2: GSE217504 and GSE175779. Single-cell RNA-seq data of CD4^+^ T cells were downloaded from PanglaoDB (https://panglaodb.se/): SRS3034950, SRS3034951, SRS3034952 and SRS3034953.

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
