# Peer review of "IGF2BP1—An Oncofetal RNA-Binding Protein Fuels Tumor Virus Propagation"

_viruses, 2023, doi:10.3390/v15071431_

Round 1

Reviewer 1 Report

The study discusses various virus species that have been linked to causing cancer, such as Epstein-Barr virus, Kaposi sarcoma-associated herpesvirus, human papillomavirus, hepatitis B and C virus, human T lymphotropic virus, and Merkel cell polyomavirus.

In the introduction, the author delves into the correlation between viral infections and tumor development. The author explains the vital role played by RBPs and miRNAs in cancer. Additionally, the author introduces IGF2BP1 and its impact on cancer. The author also explores the interactions between IGF2BP1 and various viruses. Nonetheless, it would be helpful to describe IGF2BP1's role in viral propagation in the context of its RNA-binding function in the introduction. Ideally, the paper should state its specific aims or objectives so the reader understands what to expect.

This R&D section is well-written and adds valuable insight into IGF2BP1's role in virus propagation. Furthermore, they may influence therapeutic interventions.

 In the discussion section, the authors briefly mention potential future directions and implications of their findings. However, I believe it would be beneficial for them to further elaborate on these points in a separate section.

After making the necessary amendments, the papers can be accepted.

Reviewer 2 Report

The manuscript is a relatively comprehensive review of the role of IGFBP1 in tumor virus infected cells.  There are a couple of things that the authors should consider and address.  First, the authors correctly point out that Merkel Cell virus causes human tumors but they do not discuss this virus at all.  If there is no information they should at least indicate that is the case.  Secondly, there was no discussion at all about other related polyomviruses.  Many of the other polyomaviruses have been extensively studied and in fact our original knowledge concerning p53 and RB were obtained from work with these viruses.  Although the other polyomaviruses do not cause human tumors that have been identified they should be at least noted.  Finally, there was no discussion of the basis for the target RNAs.  What is the evidence for RNA targeting? 

The manuscript is difficult to read in places.  It should be thoroughly edited to clean up the language. 

Reviewer 3 Report

The manuscript entitled of “IGF2BP1 - an oncofetal RNA-binding protein fuels tumor virus propagation” summarized the current progress on the interactions between IGF2BP1 and different viruses. Although it is interesting, there are several major issues to be addressed.

Major concerns:

1. In Fig. 1, the data illustrating the interaction between HCV and IGF2BP1 should be removed to Fig. 2.

2. It is quite necessary to summarize the differences and similarities in the interaction between IGF2BP1 and different viruses, the authors should disccuss them in detial and make more meaningful conclusions.

3. The authors should explain and disccuss why IGF2BP1 expression was altered in the virus setting.

4. A mechanism diagragh decipting the functional roles of IGF2BP1 in virus lifecylce would improve the readability of this review.

There are a few typos or missing words.

Round 2

Reviewer 3 Report

The authors have addressed all of my concerns with the original manuscript.